# Benchmarking Hamiltonian Simulation Using Graphical Processing Units

Anurag Ramesh[1][0009−0001−8360−8614], W. Michael Brown[2], Thomas Lubinski[3,4][0000−0002−3749−3430], and David E. Bernal Neira[1][0000−0002−8308−5016]

[1] Davidson School of Chemical Engineering, Purdue University, IN, USA
rames102@purdue.edu, dbernaln@purdue.edu
[2] NVIDIA Corporation, CA, USA
michbrown@nvidia.com
[3] QED-C Technical Advisory Committee - Standards, VA, USA
[4] Quantum Circuits Inc., CT, USA
tlubinski@quantumcircuits.com

**Abstract.** Simulating quantum systems is a foundational application in quantum computing, particularly in fields such as computational chemistry. We present our use of a scalable framework, the Quantum Economic Development Consortium (QED-C) Application-Oriented Benchmark Suite (QED-C), to evaluate the performance of quantum algorithms across various hardware platforms. A key focus is leveraging NVIDIA CUDA-Q, a powerful GPU-accelerated platform for quantum-classical hybrid programming, to benchmark Hamiltonian simulation, Quantum Fourier Transform (QFT), and Phase Estimation (PE).

We simulate a range of physical systems within HamLib [12], including the transverse field Ising, Heisenberg, and Fermi-Hubbard models, as well as molecules such as $H_2$ using Suzuki-Trotter evolution. Simulations were executed on NVIDIA GPUs, including the A100, H100, GH200, and GB200 systems, at Purdue University [7] and Lawrence Berkeley National Laboratory (LBNL) [8], as well as in collaboration with NVIDIA. CUDA-Q's `SpinOperator` formalism enabled emulation of circuits for up to 38 qubits on the LBNL cluster, with performance up to 3× faster than real quantum hardware. Strong scaling behavior is observed up to 32 GPUs, with execution times for some simulations reduced by more than 90%. For example, execution times for simulating a 33-qubit TFIM dropped from 19 s (1 GPU) to 2 s (32 GPUs).

Despite these gains, we observe classical HPC-like diminishing returns beyond 8 GPUs, due to inter-GPU communication bottlenecks. This impact is mitigated in the latest GB200 clusters that support extending the high-bandwidth NVLink GPU interconnect across multiple nodes. CUDA-Q proves especially effective for sampling-heavy workloads, offering near-linear scaling and improved parallel efficiency for PE and QFT. Our findings demonstrate that GPU-accelerated quantum Hamiltonian simulation with CUDA-Q provides a robust and high-throughput alternative to noisy intermediate-scale quantum (NISQ) devices, paving the way for future kernel-level optimizations and distributed quantum computing strategies.

**Keywords:** Hamiltonian simulation · CUDA-Q · GPU acceleration · Trotterization · multi-GPU scaling

## 1   Introduction

Recent advances in quantum computing have necessitated robust simulation platforms to evaluate algorithmic performance on classical hardware. Many quantum computing applications require simulating the Hamiltonians of physical and chemical systems. Hamiltonian simulation models how these quantum systems evolve over time under a specific unitary operator. Since it underpins algorithms such as Quantum Phase Estimation and is essential for solving physical problems, efficient Hamiltonian simulation serves as a crucial benchmark for evaluating quantum platforms. CUDA-Q by NVIDIA provides a platform for classically simulating these quantum Hamiltonians in the form of quantum circuits using GPU acceleration, enabling scalable testing of quantum algorithms.

Building on hardware acceleration, Dente *et al.* [2] implement a $4^{th}$ order Trotter-Suzuki decomposition to simulate nonequilibrium dynamics of interacting quantum spin systems, achieving multi-hundred-fold speedups on GPUs compared to CPUs and scaling up to 27 spins limited by GPU memory. Wittek and Calderaro [13] extend this approach to handle a wider range of physics problems—including periodic boundary conditions, many-body non-interacting particles, arbitrary stationary potentials, and imaginary time evolution—with scalability demonstrated from single-node multicore to clusters of 64+ processors and memory usage from 5 MB to 512 GB, enabling larger and more diverse simulations.

Kawase and Fuji  [4] propose accelerating classical quantum Hamiltonian simulation by clustering commuting Pauli terms and applying Clifford transformations for simultaneous diagonalization, significantly reducing memory overhead and enhancing GPU parallelism, with benchmarks showing reduction of simulation time on a 30-qubit transverse Ising model from 7.9 hours on CPU to 4.2 minutes on GPU. Faj *et al.* [3] evaluate GPU and multi-GPU acceleration in Qiskit Aer simulators, demonstrating up to $14\times$ speedup over CPU baselines and additional gains from Nvidia's cuQuantum backend, while identifying host–GPU data transfer as a scalability bottleneck. Ma and Li [6] address accurate runtime prediction for quantum programs using a graph transformer model trained on 1,510 benchmark circuits, achieving over 95% $R^2$ accuracy for simulators and 90% for hardware, surpassing existing platform estimates and revealing key runtime predictors such as two-qubit gate counts.

In this work, a series of controlled experiments were conducted to explore GPU-accelerated quantum simulation in several dimensions. We examined the scaling limits observed with currently available GPU systems, as well as the impact of load distribution across nodes on simulation execution time. The experiments were carried out on multiple generations and configurations of NVIDIA GPU systems.

The workloads specifically target simulations of Hamiltonian evolution using various models from the HamLib library [12], alongside well-established quantum algorithms such as Quantum Fourier Transform (QFT) and Quantum Phase Estimation (QPE). QFT serves as a key subroutine in many quantum algorithms, enabling efficient transformation between computational and frequency domains. QPE, in turn, builds on QFT to estimate the eigenvalues of unitary operators, making it essential for extracting spectral properties of quantum systems. These baseline algorithms serve as performance benchmarks, providing a comparative framework to assess the computational complexity of the Hamiltonian evolution operators. As our results demonstrate, Hamiltonian simulations exhibit approximately $10\times$ greater computational cost in both execution time and circuit depth compared to these foundational quantum algorithms. Performance analysis was primarily performed using the expected execution time, normalized by the maximum number of GPUs used during the run. We utilized the benchmarking framework provided by the open-source QED-C suite (Link to repository) of Application-Oriented Performance Benchmarks for Quantum Computing [10,11,5]

### 1.1   Contributions

We present multi-GPU performance benchmarks of Hamiltonian simulations using CUDA-Q across NVIDIA's A100, H100, GH200, and GB200 platforms, demonstrating strong scaling up to 256 GPUs and the ability to simulate systems of up to 40 qubits. Our study highlights the performance differences between quantum algorithms, QFT and PE, and showcases the architectural advantages using various Nvidia GPU systems. We observe significant reductions in execution time for quantum circuit simulations with increasing GPU counts and improved interconnects. These results emphasize the importance of optimized communication strategies for scalability and suggest that architectural tuning and gate fusion parameters beyond the CUDA-Q defaults can yield further performance improvements. We note that these results could serve as valuable starting points for future work on Hamiltonian simulations using GPUs.

## 2   Methods

A representative set of quantum programming tasks was selected from the open-source QED-C suite of Application-Oriented Performance Benchmarks for Quantum Computing [11,5]. The suite provides an implementation of key quantum algorithms and common application tasks structured as benchmarking problems to evaluate multiple aspects of performance in quantum computation, such as the quality of the result, the execution speed and the utilization of resources.

The tasks selected for our work fall into two categories. The first is the execution of single quantum kernels of standard quantum algorithms, such as *Quantum Fourier Transform* (QFT) or *Quantum Phase Estimation* (PE). The

second is the computation of the expectation values after simulation of a qubit-encoded quantum *Hamiltonian evolution* [1,9]. The QED-C suite uses a publicly available library of Hamiltonian problem instances, HamLib [12], as its source. HamLib is a comprehensive dataset of quantum Hamiltonians, encompassing problem sizes ranging from 2 to 1000 qubits and organized into several high-level categories, such as optimization, condensed matter, and chemistry models.

In the first category, for each of the two quantum algorithms, the test involves generating a set of quantum kernels of varying sizes, ranging from 4 to 40 qubits. The kernel instances are executed in sequence on a target GPU-accelerated quantum simulator, while performance metrics are collected and stored in a database for analysis. The key metrics recorded are listed below. (Result fidelity metrics are also computed, but are typically relevant only for physical quantum hardware devices.)

– GPU utilization (%)
– GPU execution time (s)
– GPU memory usage (MiB)

GPU utilization (%) indicates the percentage of time the GPU cores are actively engaged in computations. GPU execution time (s) refers to the total duration a specific task runs on the GPU. GPU memory usage (MiB) measures the amount of memory consumed during the execution of these tasks. In each of these tests, the execution time corresponding specifically to the GPU kernel execution time is plotted against the number of qubits, providing a direct measure of computational scalability with respect to problem size

For the Hamiltonian simulation tests, six representative Hamiltonian problems were selected. The Hamiltonian benchmark test is more involved than the simple algorithm test described above. For each, the test sweeps over problem sizes ranging from 2 to 38 qubits and generates a quantum kernel that encodes the terms of the Hamiltonian into a single Trotter-Suzuki step, using the CUDA-Q Spin Operator, and initializes the kernel to a random quantum state. After execution, the energy expectation value for that Hamiltonian is computed using the CUDA-Q `observe()` method. The same metrics shown in the list above are relevant for these tests (with the quality of expectation computation relevant primarily for quantum hardware devices, not discussed here).

All of the quantum simulations were executed across multiple high-performance computing platforms. GPU-accelerated computations used NVIDIA A100 tensor core GPUs on the Purdue Anvil Cluster and Perlmutter at Lawrence Berkeley National Laboratory (LBNL) [8], with additional evaluations performed on H100 GPUs available through the Purdue Gautschi Cluster [7]. Performance assessments also incorporated NVIDIA's latest GB200 and GH200 GPU architectures. CPU-based calculations employed AMD EPYC processors: 32-core EPYC 7543 processors on Anvil and 96-core EPYC 9654 processors on Gautschi.

The computational framework was implemented using NVIDIA CUDA Quantum (CUDA-Q), with version 0.9.1 deployed on the Anvil system and version

0.10.0 on Gautschi and Perlmutter. The Anvil and Gautschi platforms operated on Rocky Linux distributions: version 8.10 (Green Obsidian) for Anvil and version 9.4 (Blue Onyx) for Gautschi.

# 3  Results

The experimental results for multi-GPU quantum circuit simulations on A100, H100, GH200, and GB200 reveal important insights into the scaling behavior and efficiency of quantum algorithms, particularly the Quantum Fourier Transform, Phase Estimation, and HamLib, which provides a comprehensive dataset of qubit-based quantum Hamiltonians for simulation.

**Table 1.** Scaling Performance of Diverse Hamiltonian Simulations on H100 GPUs

| Hamiltonian | GPU Model | #GPUs | #Nodes | Max Qubits | Expected Time (s) |
|---|---|---|---|---|---|
| TFIM | H100 80 GB | 8 | 1 | 34 | 9.401 |
| Heisenberg | H100 80 GB | 8 | 1 | 34 | 16.575 |
| Fermi-Hubbard | H100 80 GB | 8 | 1 | 32 | 4.904 |
| Bose-Hubbard | H100 80 GB | 8 | 1 | 32 | 17.132 |
| Max3Sat | H100 80 GB | 8 | 1 | 34 | 27.936 |
| Hydrogen molecule ($H_2$) | H100 80 GB | 8 | 1 | 20 | 3.248 |
| TFIM | H100 80 GB | 4 | 1 | 34 | 13.159 |
| Heisenberg | H100 80 GB | 4 | 1 | 34 | 24.133 |
| Fermi-Hubbard | H100 80 GB | 4 | 1 | 32 | 6.061 |
| Bose-Hubbard | H100 80 GB | 4 | 1 | 32 | 23.685 |
| Max3Sat | H100 80 GB | 4 | 1 | 34 | 44.024 |
| Hydrogen molecule ($H_2$) | H100 80 GB | 4 | 1 | 20 | 3.342 |

## 3.1  Multi-GPU Scaling Behavior

The data in Table 1 showcase the expected strong scaling properties as the number of GPUs increases, where Hamiltonian simulations on 4 and 8 H100 GPUs show clear performance improvements, particularly for larger and more computationally intensive problems such as Max3Sat and Bose-Hubbard. However, for smaller systems such as $H_2$, the gains from increasing GPU count are minimal, suggesting that inter-GPU communication overhead can outweigh the benefits of parallelism for problems with limited computational granularity. These observations highlight the importance of balancing problem size, GPU count, and interconnect architecture when scaling quantum simulations on modern GPU clusters. In Figure 1, the normalized execution times (Execution time / Maximum number of GPUs used) for the HamLib Hamiltonian simulation approximately halve when doubling the GPU count from 1 to 2, 2 to 4, and 4 to 8, demonstrating an effective workload distribution across multiple GPUs.

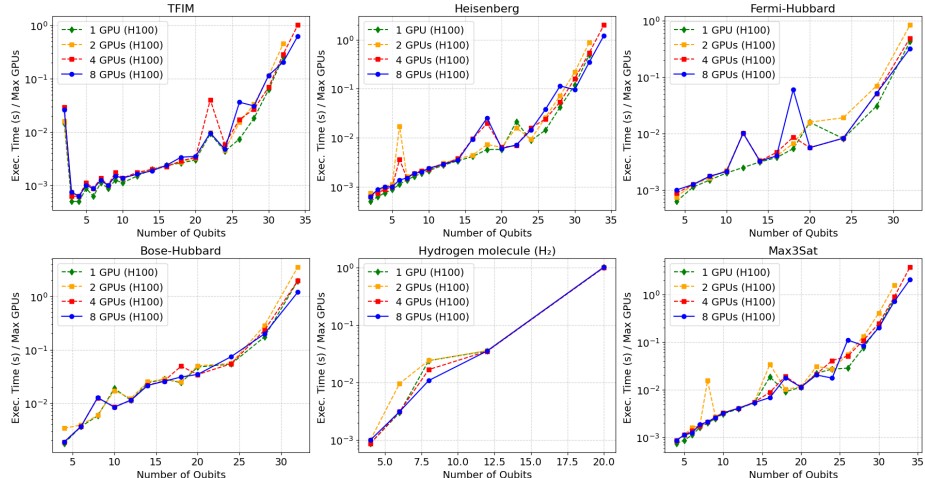

**Fig. 1.** Normalized execution time profiles for different Hamiltonians using H100 GPUs.

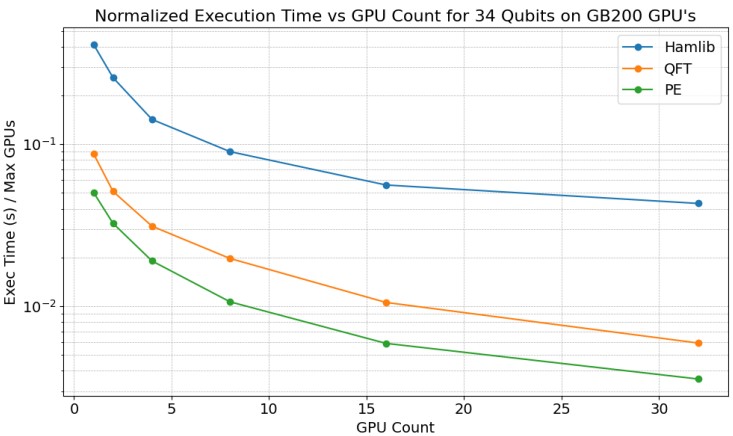

**Fig. 2.** Normalized execution time profiles for simulating 34 qubits using different GPU counts on GB200 (FP32)

Simulations executed on more than 8 GPUs experience diminishing gains in performance, indicating a drop in parallel efficiency. This reduction is primarily attributed to the higher data movement overhead across lower-bandwidth interconnects between GPUs compared to the higher-bandwidth memory hierarchy within a single GPU.

This trend is evident in the plot (Figure 2) for GB200 (34-Qubit, FP32 configuration), where there is minimal reduction in execution time as the number of GPUs increases beyond 25, illustrating the classical HPC behavior of diminishing returns in strong scaling.

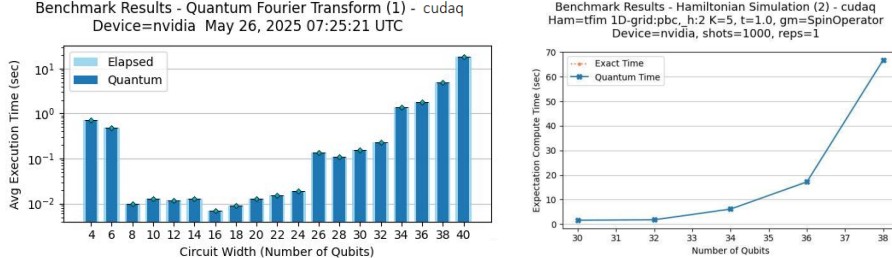

**Fig. 3.** Execution time profile for QFT algorithm and TFIM Hamiltonian evolution at increasing problem size, using 256 Perlmutter A100 GPUs. The memory load is distributed across the GPUs, enabling these simulations to be performed up to 40 or 38 qubits. Execution time grows as expected, by a factor of 4 for each increase of 2 qubits in problem size. Execution times for the Hamiltonian simulation are approximately one order of magnitude larger than those for the QFT, a consequence of the circuit depth and the cost of observable estimation. (Data collected on NERSC Perlmutter GPU partition)

In Figure 3, the execution time profile for the QFT algorithm and TFIM Hamiltonian evolution, executed on 256 Perlmutter A100 GPUs, illustrates that these simulations can be performed up to 40 or 38 qubits, respectively. The quantum simulation memory load is distributed across all GPUs, enabling execution at such a scale. However, execution time grows as expected, by a factor of 4 for each increase of 2 qubits in problem size.

The data shown in Figures 1 and 2 highlight the incremental reduction in execution time that can be achieved when using only a small number of GPUs (1, 2, 4, 8) to simulate problems of up to 34-35 qubits. In contrast, Figure 3 emphasizes the benefit gained from distributing the load over a larger number of GPUs, which significantly increases the total number of qubits that can be simulated, in this case, 38-40.

### 3.2 Performance Analysis

**Quantum Fourier Transform:** The Quantum Fourier Transform (QFT) exhibits strong scaling performance on Hopper GPUs (H100, GH200), efficiently handling large substate vectors, and achieving faster execution times as the number of GPUs increases. However, scaling benefits begin to taper off beyond 8 GPUs due to communication overhead, a common challenge in classical high-performance computing where inter-GPU communication starts to limit performance. At 32 GPUs, parallel distribution of workload deviates significantly from ideal linear scaling, highlighting data movement between GPUs as the main bottleneck (Table 2). Additionally, while initialization overhead is noticeable at lower qubit counts, it becomes less significant as simulations scale up to 38 qubits, suggesting improved amortization of startup costs at larger problem sizes.

**Phase Estimation:** Phase estimation (PE) on GB200 GPUs demonstrates significant benefits from multi-GPU scaling, with execution time decreasing from 0.804 seconds on a single GPU to 0.067 seconds on 32 GPUs (Table 3). This near-linear reduction underscores the efficiency of distributed quantum state manipulation in PE when effectively parallelized. Although communication overhead increases slightly as the number of GPUs grows, PE maintains better scaling efficiency compared to the Quantum Fourier Transform (QFT), likely due to its comparatively smaller data-movement requirements during gate operations. The optimized memory design of the Blackwell architecture for PE further facilitates a robust multi-GPU distribution, indicating that it is well suited for highly parallel quantum computing tasks.

**HamLib:** The A100 GPUs demonstrated excellent scaling performance for simulations involving up to 32 qubits on a single node, utilizing only 2 GPUs per node. However, simulating 34 qubits required the use of two nodes with a total of 16 A100 GPUs on Perlmutter, as well as 8 H100 GPUs on a single node of the Gautschi cluster. HamLib demonstrated significant scaling improvements, with execution time decreasing from 19.378 seconds on a single GPU to just 2.026 seconds on 32 GPUs, a reduction exceeding 90% (Table 3). Further scaling efforts showed intensive GPU utilization; for instance, simulating 38 qubits for the TFIM model required 16 nodes and 256 A100 GPUs on Perlmutter.

Despite substantial performance gains, the results followed the typical diminishing returns curve observed in strong-scaling high-performance computing tasks, where data transfer between GPUs limits ideal linear scaling. Overall, the findings indicate that HamLib is highly suitable for distributed execution, although further optimization of interconnect usage could enhance its efficiency at larger scales.

**Table 2.** Expected execution time for QFT simulation on GH200 with Infiniband and GB200 with multi-node NVLink

| | | QFT Time (s) | |
| --- | --- | --- | --- |
| **GPUs (GH200)** | **Qubits** | GH200 | GB200 |
| 1 | 33 | 2.847 | 1.25 |
| 2 | 34 | 3.903 | 1.632 |
| 4 | 35 | 7.166 | 2.005 |
| 8 | 36 | 8.723 | 2.478 |
| 16 | 37 | 9.812 | 2.595 |
| 32 | 38 | 10.284 | 2.717 |

### 3.3   Comparison with GB200 (33-Qubits)

The GB200 results highlight the advantages of the Blackwell GPU architecture in combination with the Grace CPU. In particular, Hamlib and PE demonstrate

**Table 3.** Expected execution time for QFT, PE, and HamLib simulations for 33 qubits on GH200 with Infiniband and GB200 with multi-node NVLink

| | HamLib (s) | | QFT (s) | | PE (s) | |
|---|---|---|---|---|---|---|
| **GPUs** | GH200 | GB200 | GH200 | GB200 | GH200 | GB200 |
| 1 | 41.46 | 19.378 | 2.847 | 1.25 | 2.047 | 0.804 |
| 2 | 26.579 | 11.985 | 1.961 | 0.788 | 1.525 | 0.516 |
| 4 | 22.396 | 6.977 | 1.832 | 0.501 | 1.288 | 0.307 |
| 8 | 20.571 | 4.127 | 1.123 | 0.31 | 0.892 | 0.179 |
| 16 | 11.886 | 2.629 | 0.683 | 0.172 | 0.539 | 0.1 |
| 32 | 8.114 | 2.026 | 0.433 | 0.109 | 0.365 | 0.067 |

significant reductions in execution times with GPU scaling. For instance, Hamlib sampling drops from 19.378s s with 1 GPU to just 2.026 s with 32 GPUs, reflecting over 90%-time reduction. PE follows a similar pattern, suggesting optimized data handling and execution pipelining with increased GPU resources. Moreover, QFT on GB200 exhibits robust scaling, with a reduction from 1.25 to 0.109 s, demonstrating effective interconnect utilization.

### 3.4   Implications for HPC and Quantum Simulation

These results suggest that multi-GPU configurations are highly effective for large-qubit simulations or parallelizable quantum circuits. The performance gain from increasing the GPU count from four to eight is minimal, showing only a slight improvement of a few seconds. However, beyond 8 GPUs, the efficiency loss becomes substantial, suggesting the need for optimized interconnects or communication-compressed strategies to maintain scalability. Interestingly, for certain qubit sizes, running the simulation on a single GPU can sometimes be faster than using eight GPUs. This counterintuitive behavior can arise due to the overhead associated with communication and data synchronization between multiple GPUs, which may outweigh the benefits of parallelism at smaller problem sizes or less optimized workloads. Additionally, the best performance tunable parameter for gate fusion may deviate from CUDA-Q defaults, indicating potential areas for custom optimization. Future access to systems with 576 Blackwell GPUs presents an opportunity to explore these strategies further, potentially mitigating communication bottlenecks and extending strong-scaling capabilities beyond current limitations.

The next steps involve optimizing the gate fusion parameters and exploring the expectation value computation strategies that can leverage distributed memory efficiently in multi-GPU setups. Additionally, techniques such as Pauli grouping and operator merging, as discussed in  [9], present promising avenues for reducing measurement overhead and communication costs during observable evaluation, making them valuable strategies for enhancing the efficiency of large-scale Hamiltonian simulations This could unlock further improvements in both

QFT and more complex quantum algorithms as we scale to larger qubit counts and higher GPU configurations.

## 4   Future Work

Future work will investigate the current limitations of GPU usage on A100s, analyzing why they require a larger number of GPUs compared to H100s for similar workloads. It will also examine whether A100 80 GB GPUs provides performance advantages over the A100 40 GB GPUs. Further efforts will focus on scaling the HamLib benchmark beyond 34 qubits using multi-node configurations and Message Passing Interface (MPI) on the H100 GPUs of the Gautschi cluster. In addition, fine-grained profiling with Nsight Systems will be conducted to enable kernel-level optimizations. The study will also include tests of Multi-Instance GPU (MIG) partitioning to assess performance in shared GPU scenarios.

The work described here evaluates the scaling characteristics of multiple GPUs that implement distributed parallel processing of the state vector associated with a single quantum kernel by allocating the memory load to enable larger qubit widths. A complementary future study will explore a distinctly different parallelization approach, in which the processing load is distributed across multiple GPUs by executing quantum kernels in parallel. Each quantum kernel will implement a subset of the commuting term groups comprising the complete quantum Hamiltonian, with the expectation value aggregated across the executions. This alternative distributed approach contrasts with our current memory distribution method, potentially offering additional performance improvements for quantum simulations involving large, complex Hamiltonians with numerous individual terms.

## Acknowledgments

We gratefully acknowledge Purdue University and the Rosen Center for Advanced Computing [7] for providing access to and computing time on the Anvil and Gautschi systems. This material is based upon work supported by the Center for Quantum Technologies under the Industry-University Cooperative Research Center Program at the US National Science Foundation under Grant No. 2224960.

The authors also thank the Quantum Economic Development Consortium (QED-C) for access to the QED-C suite of Application-Oriented Performance Benchmarks for Quantum Computing.

This research used resources of the National Energy Research Scientific Computing Center (NERSC), a U.S. Department of Energy Office of Science User Facility located at Lawrence Berkeley National Laboratory, operated under Contract No. DE-AC02-05CH11231 using NERSC award m4976. Computations were performed on the GPU partition of the Perlmutter supercomputer.

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
