# OpenReview forum: "Benchmarking Hamiltonian Simulation Using Graphical Processing Units"
_purdue.edu/Purdue_University/PQAI/2025/Symposium — PQAI 2025 Oral_

### Official Review · Reviewer_QS8Q · 2025-07-23
**This work presents a thorough benchmark of multi-GPU Hamiltonian simulation using NVIDIA's CUDA-Q framework across A100/H100/GB200 systems.**

**Rating:** 7
**Confidence:** 4

**Review:**

This paper presents a thorough benchmarking study of GPU-accelerated quantum Hamiltonian simulation using NVIDIA's CUDA-Q platform across multiple GPU architectures (A100, H100, GH200, GB200).

Strengths:

Comprehensive experimental design spanning multiple GPU architectures and problem sizes (up to 38 qubits).

Solid technical execution with clear methodology using the QED-C benchmarking framework.

Valuable practical insights about GPU scaling behavior, including identification of communication bottlenecks beyond 8 GPUs.

Good coverage of different Hamiltonian models.

Weaknesses:

Limited algorithmic novelty: primarily a performance evaluation rather than methodological contribution

Results largely confirm expected HPC scaling behavior (diminishing returns, communication bottlenecks)

Missing comparison with other quantum simulation frameworks beyond CUDA-Q

---

### Official Review · Reviewer_Cf5C · 2025-07-23
**Benchmarking Hamiltonian Simulation Using Graphical Processing Units**

**Rating:** 7
**Confidence:** 3

**Review:**

Recommended: Oral presentation

Good points
This paper tackles the practically important topic of large-scale benchmarking of quantum circuit simulations using GPUs, and I think that both the comprehensiveness of the experiments and the analysis of the results are at a high level. Although no new algorithms are proposed, the paper has academic value in that it significantly scales up existing research by making full use of the latest GPU platform (CUDA-Q). In particular, the results, such as the simulation demonstration of up to 256 GPUs and 40 qubits and the quantitative evaluation of communication bottlenecks, provide useful knowledge for the development of future quantum computing simulation methods. The benchmarking method is also standard and highly reliable.

Points that need improvement
However, in terms of academic impact, due to the nature of the paper being "demonstration research on acceleration using GPUs," emphasis is placed on the report of performance evaluation rather than on novel conceptual proposals, and the theoretical novelty is somewhat limited.

---

### Decision · Program_Chairs · 2025-07-29

Accept (Oral)